# Synthetic Data is Sufficient for Zero-Shot Visual Generalization from Offline Data

**Ahmet H. Güzel**                                            *ahmet.guzel.23@ucl.ac.uk*
*University College London AI Centre*

**Ilija Bogunovic**                                              *i.bogunovic@ucl.ac.uk*
*University College London AI Centre*

**Jack Parker-Holder**                                    *j.parker-holder@ucl.ac.uk*
*University College London AI Centre*

**Reviewed on OpenReview:** *https://openreview.net/forum?id=gFmSFa408D*

## Abstract

Offline reinforcement learning (RL) offers a promising framework for training agents using pre-collected datasets without the need for further environment interaction. However, policies trained on offline data often struggle to generalise due to limited exposure to diverse states.The complexity of visual data introduces additional challenges such as noise, distractions, and spurious correlations, which can misguide the policy and increase the risk of overfitting if the training data is not sufficiently diverse. Indeed, this makes it challenging to leverage vision-based offline data in training robust agents that can generalize to unseen environments. To solve this problem, we propose a simple approach—generating additional synthetic training data. We propose a two-step process, first *augmenting* the originally collected offline data to improve zero-shot generalization by introducing diversity, then using a diffusion model to *generate* additional data in latent space. We test our method across both continuous action spaces (Visual D4RL) and discrete action spaces (Procgen), demonstrating that it significantly improves generalization without requiring any algorithmic changes to existing model-free offline RL methods. We show that our method not only increases the diversity of the training data but also significantly reduces the generalization gap at test time while maintaining computational efficiency. We believe this approach could fuel additional progress in generating synthetic data to train more general agents in the future.

## 1 Introduction

Offline reinforcement learning (RL) offers a compelling approach for training agents using pre-collected datasets without additional environment interaction (Levine et al., 2020). This paradigm is particularly valuable in domains like healthcare (Liu et al., 2020), robotics (Singla et al., 2021), and autonomous driving (Kiran et al., 2021), where real-time data collection can be costly or risky. However, generalizing policies trained on high-dimensional visual inputs in offline RL remains a significant challenge, and it has received relatively little attention in the research community. Agents may learn irrelevant correlations between visual features and actions, reducing their ability to perform well in new settings (Song et al., 2019; Raileanu & Fergus, 2021). Additionally, offline RL policies tend to exhibit risk-averse behavior, avoiding novel actions in unfamiliar states, which further hampers generalization (Mediratta et al., 2024). To tackle these challenges, we propose a two-step method that combines data augmentation with diffusion model-based upsampling to improve generalization in offline RL. While both data augmentation (Laskin et al., 2020; Yarats et al., 2021a;b; Raileanu et al., 2021) and the use of diffusion models for replay buffer upsampling (Lu et al., 2023b) have been explored independently in the online RL domain, our contribution lies in their integration and adaptation for the unique challenges of offline RL to achieve greater diversity and more robust generalization.

First, we apply data augmentation techniques to the offline dataset, introducing variability that helps reduce overfitting. Then, we use a diffusion model to upsample the augmented dataset in the latent space, generating additional synthetic data points that capture unseen transitions. This approach broadens the distribution of experience replay data without incurring significant computational overhead, allowing policies to generalize more effectively to new environments.Our results demonstrate that our method significantly reduces the generalization gap across various difficulty levels in two recent visual offline RL benchmarks, highlighting the effectiveness of combining augmentation and diffusion-based upsampling.

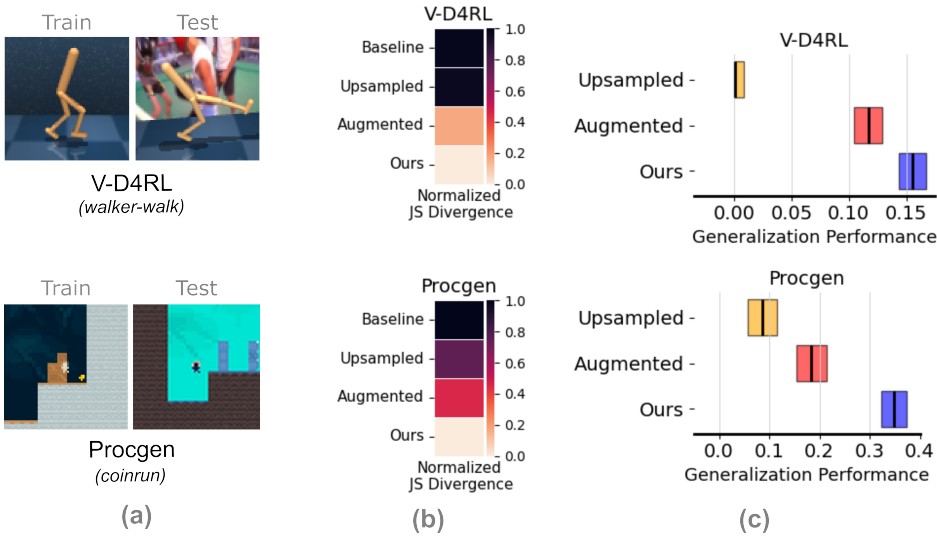

Figure 1: V-D4RL (continuous) and Procgen (discrete) benchmarks illustrate the generalization challenge in offline RL by showcasing visual differences between training and testing environments (a). Jensen-Shannon divergence heatmaps demonstrate how well each method aligns training and testing distributions, with our two-stage approach outperforming the data upsampling, and augmentation methods; darker colors indicate higher divergence (b). The performance of each method in unseen environments, normalized to the baseline, consistently shows our approach performing best with reduced variability across runs (c). For detailed analysis, refer to Section 5 for details.

To summarize, our contributions are:

- We introduce a practical method that integrates data augmentation and diffusion-based upsampling to improve generalization in offline RL from visual inputs, *without requiring modifications* to existing model-free offline RL algorithms.
- We show our approach expands data diversity without increasing computational costs, improving zero-shot generalization across both continuous and discrete control tasks.
- To the best of our knowledge, we are the first to propose a practical, scalable method that addresses generalization in both continuous (V-D4RL) and discrete (Procgen) control tasks within offline RL.

## 2 Background

### 2.1 Offline Reinforcement Learning from Visual Observations

Reinforcement learning (RL) typically involves an agent interacting with an environment modeled as a Markov Decision Process (MDP) (Sutton & Barto, 2018), where the objective is to optimize the expected cumulative return $J(\pi) = \mathbb{E}_{\pi,P,\rho_0} \left[ \sum_{t=0}^{\infty} \gamma^t R(s_t, a_t) \right]$, where $J(\pi)$ represents the expected return of a policy $\pi$, $\mathbb{E}_{\pi,P,\rho_0}$ is the expectation over the policy $\pi$, the environment dynamics $P(s'|s,a)$, and the initial state distribution $\rho_0$. The term $\gamma \in [0,1)$ is the discount factor, controlling how much future rewards are valued, and $R(s_t, a_t)$ is the reward function at time step $t$, depending on the state $s_t$ and action $a_t$. The goal is to find a policy $\pi$ that maximizes this cumulative discounted return over time.

However, in offline RL, the agent must learn from a static dataset $D = \{(o_i, a_i, r_i, o_i')\}_{i=1}^N$, without any interaction with the environment during training (Levine et al., 2020). This dataset observations collected by one or more behavior policies. When dealing with visual observations (high-dimensional inputs), additional challenges arise.Unlike proprioceptive observations in standard RL, visual inputs are prone to noise and spurious correlations (Lu et al., 2023a), making offline RL particularly vulnerable to overfitting. Small environmental changes (e.g. lighting or background) can cause significant shifts in data distribution. Without interaction to correct for these shifts, agents struggle to generalize from visual observations (Raileanu & Fergus, 2021). Given these challenges, the core problem is: *How can we improve the generalization performance of model-free offline RL methods from visual observations and ensure the robust deployment of agents in unseen environments for both continuous and discrete action spaces?*

## 2.2 Diffusion Models

Diffusion models generate data by reversing a noise-adding process, starting from noise and gradually denoising to recover the original data distribution (Ho et al., 2020; Rombach et al., 2022). Noise removal is guided by a learned denoising model $D_\theta(x; \sigma)$, trained using an L2 objective:

$$\min_\theta \mathbb{E}_{x \sim p, \sigma, \epsilon \sim \mathcal{N}(0, \sigma^2 I)} \|D_\theta(x + \epsilon; \sigma) - x\|_2^2. \tag{1}$$

This allows the model to estimate the data distribution at different noise levels. Additional details, including the use of ODEs or SDEs for the reverse process, can be found in Karras et al. (2022). Diffusion models have shown superior performance in generating diverse synthetic datasets compared to Generative Adversarial Networks (GAN) and Variational Autoencoders (VAEs), which makes them particularly effective for improving generalization in reinforcement learning Lu et al. (2023b). This is why we chose diffusion models, as their ability to generate diverse data makes them ideal for our approach.

## 3 Method

### 3.1 Overview of the Proposed Method

To address the generalization challenges of off-line RL from visual observations, we present a simple, practical approach that combines data augmentation and diffusion model-based upsampling.

1. **Data Augmentation to Increase Initial Dataset Diversity:** We apply specific data augmentation techniques to offline datasets to increase the diversity of the initial dataset $\mathcal{D}_0$. This step aims to introduce variability and reduce overfitting to spurious correlations in visual inputs.
2. **Upsampling with Diffusion Models:** We employ diffusion model to upsample (SynthER (Lu et al., 2023b)) the augmented dataset $\mathcal{D}_0$, generating additional synthetic samples in the latent space. This further increases dataset diversity and helps the policy generalize better to unseen environments.

By integrating data augmentation with diffusion model-based upsampling, our method effectively covers a wider range of potential scenarios without significantly increasing computational overhead.

### 3.2 Step 1: Data Augmentation for Initial Dataset Diversity Enchancement

To construct an initial dataset $\mathcal{D}_0$ that captures key environment dynamics, we apply a set of data augmentation techniques to improve robustness to variations in visual inputs. Specifically, we focus on *rotation*, *color jittering*, *color cutout*, and *background image overlay*, which were empirically found to improve generalization. These augmentations introduce variations that prevent the agent from learning spurious correlations in the visual inputs. However, we believe that data augmentation alone may not fully capture the diversity of real-world scenarios, particularly in unseen environments, making it necessary to complement this with

synthetic data generation through diffusion models, which has proven to improve diversity even more effectively than augmentation techniques (Lu et al., 2023b). Full details for this part are given in supplementary material.

### 3.3 Step 2: Latent Space Upsampling with Diffusion Models

We first train an encoder-based model-free visual offline RL algorithm on the augmented dataset $\mathcal{D}_0$, using the selected image augmentation techniques described in Section **??**. For the V-D4RL benchmark, we use the **DrQ+BC** algorithm (Lu et al., 2023a), while **CQL** is employed for the Procgen benchmark (Kumar et al., 2020). In both cases, the networks consist of a CNN encoder $f_\xi$, policy network $\pi_\phi$, and Q-function networks $Q_\theta$. This initial training enables the model to learn robust representations from diverse visual inputs, tailored to the specific requirements of each environment.

After the initial training, we extract latent space parameters from the augmented dataset by passing the augmented observations through the trained encoder $f_\xi$ and the linear head layers, which sit between the encoder and the MLPs of the policy and Q-function networks. For each transition $(s, a, r, s')$ in $\mathcal{D}_0$, the following computations are made:

$$h = f_\xi(\text{Augment}(s)), \quad h' = f_\xi(\text{Augment}(s')) \tag{2}$$

$$z_\pi = \pi_\phi^{\text{lin}}(h), \quad z'_\pi = \pi_\phi^{\text{lin}}(h') \tag{3}$$

$$z_Q = Q_\theta^{\text{lin}}(h), \quad z'_Q = Q_\theta^{\text{lin}}(h') \tag{4}$$

where $\pi_\phi^{\text{lin}}$ and $Q_\theta^{\text{lin}}$ denote the linear head layers of the policy and Q-function networks.

We combine the latent representations to construct the latent transitions using concatenation, which produces feature vectors with dimensions equal to the sum of the individual vectors' dimensions:

$$z = z_\pi \cdot z_Q, \quad z' = z'_\pi \cdot z'_Q \tag{5}$$

resulting in the latent dataset $\mathcal{D}_{\text{latent}} = \{(z, a, r, z')\}$, which is used to train the diffusion model $\mathcal{M}_{\text{diff}}$. Following Lu et al. (2023b) and Karras et al. (2022), the diffusion model generates synthetic latent transitions $(z_d, a_d, r_d, z'_d)$, producing the upsampled dataset $\mathcal{D}_{\text{diff}}$.

Finally, we combine the original and upsampled datasets to create an expanded dataset:

$$\mathcal{D}_{\text{ups}} = \mathcal{D}_{\text{latent}} \cup \mathcal{D}_{\text{diff}} \tag{6}$$

The encoder $f_\xi$ and the linear head layers of the policy and Q-functions are frozen during fine-tuning, allowing the training to focus on refining the MLP layers of the policy and value networks using the diverse data provided by $\mathcal{D}_{\text{ups}}$. This ensures stable representations while improving the model's ability to generalize to unseen environments with minimal computational overhead. Our empirical approach, which combines data augmentation with synthetic data generation through upsampling in the latent space, significantly increases the diversity of the data set, as demonstrated in our results (Section 5). Figure 2 illustrates the architecture of our method, built on the DrQ+BC model.

## 4 Experimental Setup

### 4.1 Environments and Datasets

We evaluated our method on two challenging offline RL benchmarks that test generalization capabilities in different domains:

- **Visual D4RL (V-D4RL)** (Lu et al., 2023a): This benchmark is a visual input version of the D4RL benchmark (Fu et al., 2021) and focuses on continuous control tasks with visual input. It features varying levels of visual distractions (easy, medium, hard) and is designed to assess generalization in continuous action spaces.

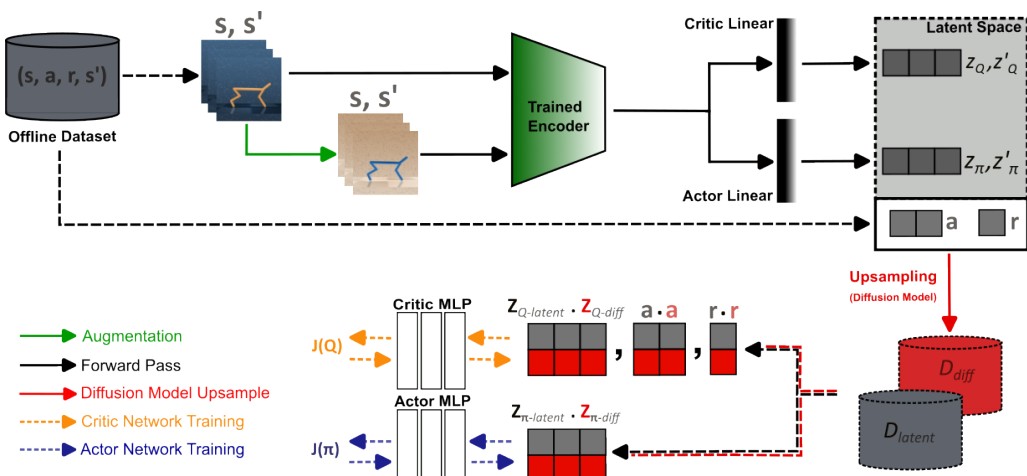

Figure 2: Illustration of our method in the V-D4RL benchmark using the DrQ+BC network. Green arrows indicate data augmentation, while blue and orange arrows represent the training flows for the actor and critic networks, respectively. The red components highlight the diffusion model upsampling process, which generates additional latent space transitions to increase the dataset diversity.

- **Offline Procgen** (Mediratta et al., 2024): This is an offline version of Procgen benchmark Cobbe et al. (2020) procedurally generated games that targets discrete control tasks. It tests zero-shot generalization to entirely unseen levels.

These offline RL benchmarks were chosen to comprehensively evaluate our method's performance across diverse environments, including both continuous and discrete action spaces, as well as its ability to generalize in the presence of visual distractions and to completely novel scenarios. Figure 3 illustrates sample observations from the V-D4RL and Procgen datasets, highlighting the visual diversity and complexity across training and testing environments.

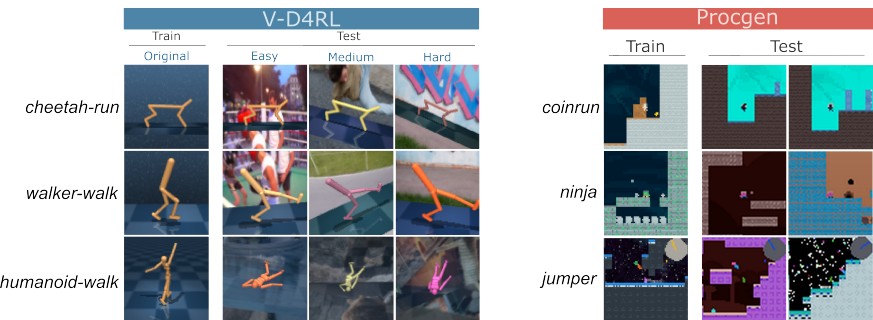

Figure 3: Sample screenshots from the V-D4RL (left) and Procgen (right) datasets, showing the training environments and testing environments.

For our experiments, we generated different dataset variants to evaluate the effectiveness of our method. We used a *Baseline* dataset without augmentation or upsampling, which was originally provided by both benchmarks; an *Upsampled* dataset where the original dataset was increased in size using diffusion model-based upsampling, without any augmentation; an *Augmented* dataset where data augmentation techniques were applied without changing the dataset size; and an **Ours** dataset that combined both augmentation and upsampling. For more details on the experimental setup, please refer to the supplementary material.

### 4.2 Implementation Details

For VD4RL we used the **DrQ+BC** (Lu et al., 2023a) algorithm, which extends DrQ-v2 (Yarats et al., 2021a), using BC (as in TD3BC, Fujimoto & Gu (2021)). For Procgen we used **Conservative Q-Learning (CQL)** (Kumar et al., 2020). The hyperparameters, network architectures, and other implementation details follow the standard settings provided in the original benchmark papers. For completeness, we provide all hyperparameters and network architecture details in supplementary material.

### 4.3 Evaluation Metrics

#### 4.3.1 Generalization Performance Metric

To quantify our model's ability to generalize to unseen environments with visual distractions, we adopt a generalization performance metric inspired by the normalization approach commonly used in reinforcement learning (RL) studies, such as the Procgen benchmark Cobbe et al. (2020) which defines the normalized return ($R_{\text{norm}}$) as:

$$R_{\text{norm}} = \frac{R - R_{\text{min}}}{R_{\text{max}} - R_{\text{min}}}, \tag{7}$$

where $R$ is the agent's performance, $R_{\text{min}}$ represents the lowest possible score (e.g., the baseline's test performance), and $R_{\text{max}}$ corresponds to the highest possible score. Inspired by this formulation, we define our *Generalization Performance* as the proportion of the baseline's generalization gap that our method closes. Specifically, we calculate it by subtracting the baseline's mean test return ($B_{\text{test}}$) from our method's mean test return ($T_{\text{test}}$), then dividing by the difference between the baseline's mean training return ($B_{\text{train}}$) and $B_{\text{test}}$. Generalization performance value of 1 indicates that our method completely eliminates the gap, providing a clear benchmark for comparison.

#### 4.3.2 Latent Space Distribution Analysis

To gain deeper insights into the impact of our method on learned representations, we performed a latent space distribution analysis using the Jensen-Shannon (JS) divergence (Menéndez et al., 1997). For each dataset variant, we extracted latent space representations by passing both training and testing observations (collected during the evaluation steps, as detailed in supplementary material through the trained encoder $f_\xi$ and the actor-critic networks. Let $h_{\text{train}}$ and $h_{\text{test}}$ represent the sets of latent representations for training and testing observations, respectively. For each environment, we estimated probability distributions over the latent space dimensions using kernel density estimation (KDE) to non-parametrically capture the distributions of $h_{\text{train}}$ and $h_{\text{test}}$. We then computed the JS divergence between the training and testing distributions for each dimension, resulting in a divergence vector $\mathbf{d} = [d_1, d_2, \ldots, d_n]$, where $n$ represents the dimensionality of the latent space. The mean JS divergence $\bar{d}$ across all dimensions was used to summarize how closely the training and testing distributions aligned, with lower divergence indicating better generalization. To facilitate comparison across different environments, we normalized the mean JS divergence values within each environment, ensuring consistency in scale. A closer match between training and test distributions suggests improved generalization performance by our method.

## 5 Experimental Results and Discussion

### 5.1 V-D4RL Benchmark Results

As seen in Figure 4, models trained on the **Baseline** and **Upsampled** datasets showed minimal improvement in the generalization gap across all environments (*cheetah-run*, *walker-walk*, and *humanoid-walk*), suggesting that upsampling alone does not significantly improve generalization. The baseline consistently shows a generalization gap of 0 because the results are normalized over it, and thus its results are not explicitly shown. In contrast, the **Augmented** model demonstrated a substantial reduction in the generalization gap,

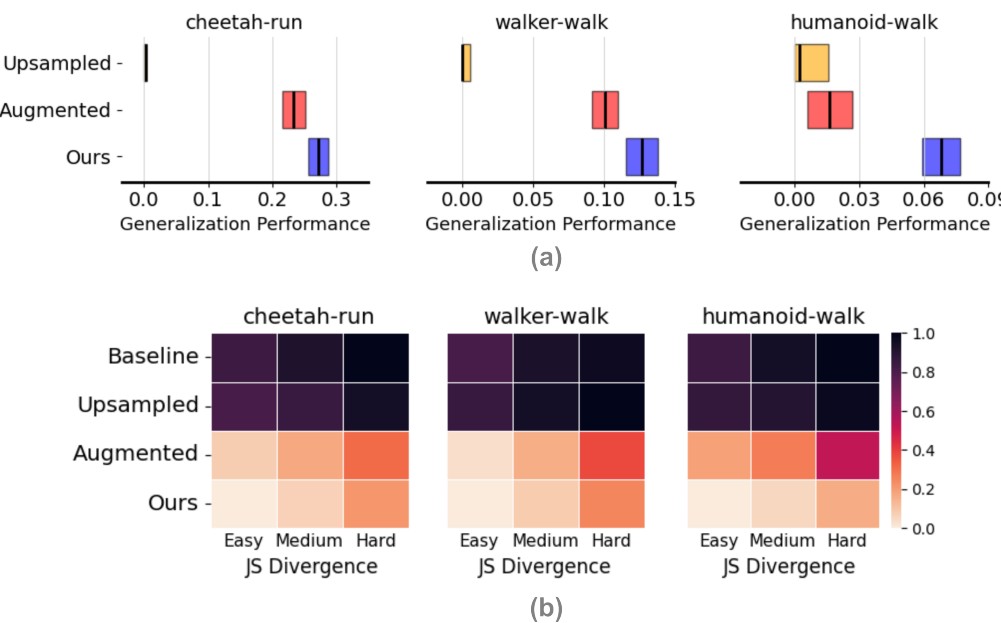

Figure 4: (a) Generalization performance averaged across all difficulty levels (easy, medium, hard) for different environments. (b) Normalized JS divergence values for each environment which normalized relative to that environment. Lower values (lighter colors) indicate a closer alignment between the distributions of training and test data, suggesting better generalization.

highlighting the importance of data augmentation in improving robustness against unseen visual perturbations. The best results were achieved with the **Ours** dataset, where combining data augmentation with upsampling further improved generalization across all environments. The JS Divergence analysis similarly showed that the **Ours** dataset achieved the lowest divergence, indicating a closer alignment between the training and testing distributions in the latent space. This further supports that combining augmentation with upsampling leads to a more consistent latent representation and, consequently, better generalization. These findings align with the generalization gap and return values in Table 1.

Table 1: Performance evaluation on the V-D4RL benchmark across different datasets and environments, trained using the DrQ+BC algorithm. All return values are based on the mean over five random seeds.

| Environment | Method | Original | Easy | Medium | Hard | Test Mean |
|---|---|---|---|---|---|---|
| *cheetah-run* | Baseline | $250.1 \pm 10.9$ | $4.2 \pm 1.3$ | $3.1 \pm 0.6$ | $3.3 \pm 0.8$ | $3.53 \pm 0.9$ |
| | Upsampled | $315.2 \pm 20.1$ | $4.6 \pm 0.7$ | $3.5 \pm 0.3$ | $4.1 \pm 0.9$ | $4.06 \pm 0.7$ |
| | Augmented | $350.5 \pm 15.5$ | $81.2 \pm 8.1$ | $60.7 \pm 6.2$ | $41.3 \pm 3.4$ | $61.1 \pm 5.9$ |
| | **Ours** | $360.2 \pm 10.1$ | $86.1 \pm 7.1$ | $71.2 \pm 6.0$ | $54.4 \pm 2.4$ | $\mathbf{70.6 \pm 5.2}$ |
| *walker-walk* | Baseline | $570.2 \pm 6.7$ | $35.4 \pm 2.4$ | $31.6 \pm 3.3$ | $29.9 \pm 2.2$ | $32.3 \pm 0.9$ |
| | Upsampled | $665.2 \pm 7.2$ | $32.4 \pm 1.4$ | $30.3 \pm 2.7$ | $28.9 \pm 1.8$ | $30.5 \pm 2.0$ |
| | Augmented | $799.5 \pm 10.5$ | $131.5 \pm 10.1$ | $75.1 \pm 5.2$ | $50.5 \pm 1.4$ | $85.7 \pm 5.6$ |
| | **Ours** | $845.9 \pm 6.1$ | $141.1 \pm 12.1$ | $92.3 \pm 2.3$ | $65.7 \pm 1.6$ | $\mathbf{99.7 \pm 5.3}$ |
| *humanoid-walk* | Baseline | $15.4 \pm 2.1$ | $1.3 \pm 0.2$ | $1.0 \pm 0.3$ | $1.1 \pm 0.2$ | $1.1 \pm 0.2$ |
| | Upsampled | $20.4 \pm 2.3$ | $1.3 \pm 0.3$ | $1.3 \pm 0.4$ | $1.1 \pm 0.1$ | $1.2 \pm 0.3$ |
| | Augmented | $25.4 \pm 2.7$ | $1.6 \pm 0.5$ | $1.5 \pm 0.3$ | $1.2 \pm 0.1$ | $1.4 \pm 0.3$ |
| | **Ours** | $28.5 \pm 1.4$ | $2.4 \pm 0.2$ | $2.3 \pm 0.1$ | $1.7 \pm 0.2$ | $\mathbf{2.2 \pm 0.2}$ |

The results indicate that data augmentation improves generalization amidst visual distractions. Our approach achieves notable generalization performance in zero-shot testing circumstances.

### 5.1.1 Leveraging Fixed Distracting Data for Improved Generalization

Building on our findings from the V-D4RL benchmark, where our two-stage approach of augmentation and upsampling significantly improved generalization, we further tested the robustness of our method. Although previous research (Lu et al., 2023a) indicated that training with fixed distracting datasets—containing hand-crafted distractions—offered minimal benefits for generalization , we hypothesized that our method could effectively leverage even a small portion of such data. To test this, we incorporated 5% of the fixed distracting data into our training dataset, combining it with 95% of the original baseline data to create a composite dataset for baseline. We then applied our augmentation and upsampled technique, we refers to this as *Ours Fixed Data Distraction (FDD)*. For more details on the experimental setup, please refer to the supplementary material.

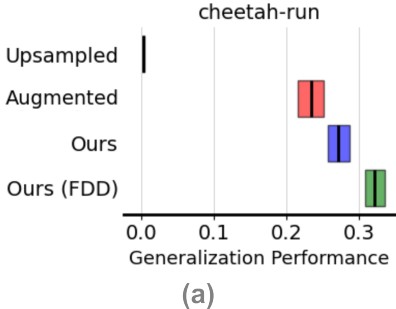
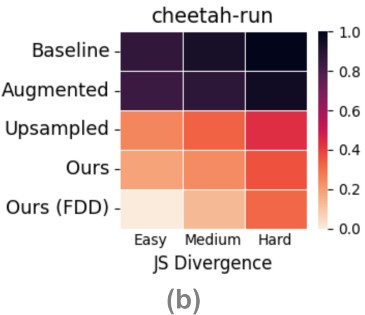

(a)           (b)

Figure 5: (a) Generalization performance averaged over all difficulty levels, and (b) comparison of normalized JS divergence values for cheetah-run expert dataset.

As illustrated in Figure 5, including just 5% of the fixed distracting data improved generalization performance across all test environments, further narrowing the generalization gap. The latent space distributions also aligned more closely with the test data, as evidenced by reduced JS divergence values. Notably, we saw significant improvements, especially on medium and hard difficulty levels, when this fixed distracting data was combined with our augmentation and upsampling techniques.(Table 2).

Table 2: Performance on the *cheetah-run* expert dataset with and without incorporating 5% fixed distracting data (FDD). Results are based on the mean of five random seeds.

| Environment | Method | Original | Easy | Medium | Hard | Test Mean |
|---|---|---|---|---|---|---|
| *cheetah-run* | Baseline | $250.1 \pm 10.9$ | $4.2 \pm 1.3$ | $3.1 \pm 0.6$ | $3.3 \pm 0.8$ | $3.53 \pm 0.9$ |
| | Upsampled | $315.2 \pm 20.1$ | $4.6 \pm 0.7$ | $3.5 \pm 0.3$ | $4.1 \pm 0.9$ | $4.06 \pm 0.7$ |
| | Augmented | $350.5 \pm 15.5$ | $81.2 \pm 8.1$ | $60.7 \pm 6.2$ | $41.3 \pm 3.4$ | $61.1 \pm 5.9$ |
| | Ours | $360.2 \pm 10.1$ | $86.1 \pm 7.1$ | $71.2 \pm 6.0$ | $54.4 \pm 2.4$ | $70.6 \pm 5.2$ |
| | **Ours (FDD)** | $267.9 \pm 7.2$ | $103.4 \pm 5.3$ | $85.4 \pm 5.0$ | $59.8 \pm 3.4$ | $\mathbf{82.8 \pm 4.6}$ |

| Environment | Method | (Test Mean) / Train | Train - (Test Mean) |
|---|---|---|---|
| *cheetah-run* | Baseline | 0.01 | 246.6 |
| | Upsampled | 0.01 | 311.1 |
| | Augmented | 0.17 | 289.4 |
| | **Ours** | 0.20 | 289.6 |
| | **Ours (FDD)** | **0.31** | **185.1** |

Our approach effectively utilizes this data to add diversity and improve robustness to unseen distractions, leading to an intriguing question:

*Could incorporating a small, strategically chosen subset of data that closely aligns with the evaluation distribution—when combined with augmentation and upsampling techniques—offer a viable strategy to improve generalization in few-shot learning scenarios?*

## 5.2 Results on Procgen Benchmark

Our results on the *Offline Procgen Benchmark* further validate the generalization capabilities of our method, demonstrating its effectiveness not only in continuous control environments like V-D4RL but also in discrete control tasks. Figure 6 illustrates the generalization performance across three different environments across datasets.

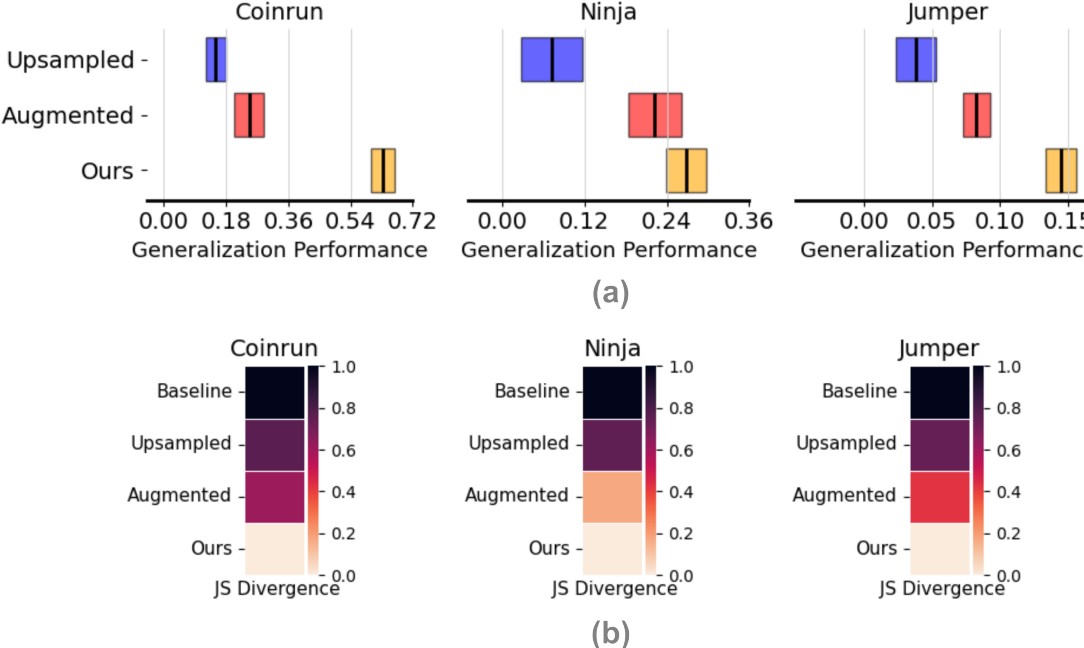

Figure 6: Performance evaluation on the Procgen benchmark across three different games, trained using the CQL algorithm. (a) Generalization performance for three games in Procgen. (b) Comparison of normalized JS divergence values.

Table 3: Performance of the CQL algorithm on the Procgen Benchmark, based on the mean over five random seeds.

| Environment | Method | Train Return | Test Return |
|---|---|---|---|
| *Coinrun* | Baseline | $8.51 \pm 0.27$ | $7.17 \pm 0.27$ |
| | Upsampled | $8.96 \pm 0.32$ | $7.37 \pm 0.37$ |
| | Augmented | $8.65 \pm 0.31$ | $7.50 \pm 0.54$ |
| | **Ours** | $8.74 \pm 0.34$ | $\mathbf{8.02 \pm 0.44}$ |
| *Ninja* | Baseline | $5.94 \pm 0.23$ | $4.41 \pm 0.21$ |
| | Upsampled | $6.18 \pm 0.39$ | $4.52 \pm 0.41$ |
| | Augmented | $5.83 \pm 0.31$ | $4.75 \pm 0.35$ |
| | **Ours** | $6.05 \pm 0.27$ | $\mathbf{4.82 \pm 0.27}$ |
| *Jumper* | Baseline | $7.62 \pm 0.19$ | $4.23 \pm 0.24$ |
| | Upsampled | $7.94 \pm 0.32$ | $4.36 \pm 0.28$ |
| | Augmented | $7.55 \pm 0.28$ | $4.51 \pm 0.19$ |
| | **Ours** | $7.35 \pm 0.24$ | $\mathbf{4.72 \pm 0.22}$ |
| **Environment** | **Method** | **Test / Train** | **Train - Test** |
| *Procgen Averaged* | Baseline | 0.72 | 2.09 |
| | Upsampled | 0.70 | 2.28 |
| | Augmented | 0.76 | 1.76 |
| | **Ours** | **0.79** | **1.53** |

As shown in Table 3, applying data augmentation alone to the baseline dataset led to marginal improvements. However, our proposed method—creating augmented and upsampled dataset—yielded the most significant improvements across all three games. These results underscore the effectiveness of our solution, confirming its ability to improve generalization performance in discrete control tasks like Procgen, in line with the gains observed in the V-D4RL continuous control benchmark. Thus, our method demonstrates its effectiveness not only in continuous control tasks but also in discrete control environments. While the improvements in Procgen are consistent with the trends observed in V-D4RL, they validate the generalization capabilities of our solution across a broader range of offline RL challenges. It is important to note that Mediratta et al. (Mediratta et al., 2024) provided an offline Procgen dataset collected from only 200 levels, limiting our ability to compare the effects of increasing the number of levels on generalization performance relative to our method. Despite this limitation, our results demonstrate that without expanding the dataset's level diversity, our approach significantly amplifies generalization performance. This suggests that our method effectively compensates for the limited number of levels through data augmentation and upsampling alone.

## 6 Related Work

**Generalization in RL** has been extensively studied, primarily in the context of online RL. A substantial body of work has focused on training agents to generalize across novel transition dynamics and reward functions (Rajeswaran et al., 2018; Machado et al., 2017; Packer et al., 2019; Cobbe et al., 2020; Kirk et al., 2023; Justesen et al., 2018; Nichol et al., 2018; Küttler et al., 2020; Bengio et al., 2020; Bertran et al., 2020; Ghosh et al., 2021; Lyu et al., 2024; Ehrenberg et al., 2022; Lyle et al., 2022; Dunion et al., 2023; Almuzairee et al., 2024). RL environments such as Procgen (Cobbe et al., 2020) and the NetHack Learning Environment (Kumar et al., 2020) have been specifically developed to assess generalization in online RL. However, these studies largely focus on interactive settings where agents can gather new data during training, leaving generalization in offline RL relatively unexplored.

**Visual offline RL** introduces additional challenges, particularly when using large-scale datasets. Datasets like Atari, StarCraft, and MineRL contain millions of samples but require significant computational resources, limiting their accessibility to many researchers (Agarwal et al., 2020; Vinyals et al., 2017; Fan et al., 2022). In contrast, benchmarks such as *V-D4RL* and *offline Procgen* (Lu et al., 2023a; Mediratta et al., 2024) offer more accessible alternatives, with 100,000 and 1 million samples per environment, respectively, while still supporting meaningful evaluation of generalization in continuous and discrete control tasks. Both benchmarks highlight the generalization challenges in offline RL, especially with model-free methods. Our work extends these efforts by evaluating generalization across diverse, procedurally generated environments in both continuous and discrete control tasks.

**Data augmentation in RL** has been widely successful in improving generalization in online RL methods (Yarats et al., 2021b;a; Raileanu et al., 2021; Laskin et al., 2020; Ma et al., 2024; Corrado et al., 2024; Pitis et al., 2020; Sinha et al., 2021; Lee et al., 2024), but its application in offline RL remains underexplored. Our work leverages augmentation to address generalization in offline settings, demonstrating its potential for visual tasks. Unlike methods such as DrAC (Raileanu et al., 2021) and SVEA (Hansen et al., 2021), which involve algorithmic changes in online RL settings, our approach focuses on non-algorithmic enhancements using simple visual augmentations, as demonstrated by (Laskin et al., 2020) combined with diffusion-based upsampling. This combination improves data diversity and generalization, providing a scalable, practical solution for offline RL settings.

**Diffusion models** have emerged as a promising tool for improving RL solutions, though they have primarily been used as policies or planners rather than as data synthesizers (Zhu et al., 2024; Jackson et al., 2024). Recent works, such as *ROSIE* (Yu et al., 2023) and *GenAug* (Chen et al., 2023), have employed diffusion models for synthetic data generation to improve generalization. However, these methods operate in online robotic learning contexts, relying on continuous interaction with the environment. In contrast, our approach applies diffusion-based upsampling in offline RL, where additional interaction with the environment is not possible. Inspired by *SynthER* (Lu et al., 2023b), our method improves data diversity through upsampling, making it the first diffusion model-based data synthesis aimed at solving the generalization problem in model-free offline RL with visual inputs, across both continuous and discrete control tasks.

## 7 Conclusion

We presented a practical two-step approach that improves generalization in offline reinforcement learning from visual inputs. By combining targeted data augmentation with diffusion model-based synthetic data generation in the latent space, our approach increases training data diversity without significant computational overhead, allowing model-free offline RL algorithms to better handle risk-averse behavior in unseen environments. Our experiments on the V-D4RL benchmark (continuous control) and Procgen benchmark (discrete control) demonstrate that our approach consistently reduces the generalization gap and improves performance in unseen environments. Additionally, our method effectively leverages small amounts of hand-crafted, fixed distracting data to further improve generalization, suggesting potential applications for few-shot learning in offline RL. These benchmarks challenge the development of better offline RL algorithms for visual observations, and to our knowledge, we are the first to apply this approach across both continuous and discrete action spaces. By broadening the data distribution in both pixel and latent spaces, we provide a scalable two-step solution to the generalization challenges in offline RL.

*While our method shows significant improvements, there are certain limitations.* Although working in the latent space keeps computational overhead relatively low, extending this approach to the pixel space would introduce significantly higher costs, especially in environments with high-resolution visual inputs. Additionally, our method required extensive experimentation to identify the best settings for data augmentation and diffusion model parameters, which may limit its immediate applicability to more complex environments like robotics or autonomous driving. We opted for model-free algorithms in this work due to their sampling efficiency and lower computational load. However, these algorithms may face limitations in handling more complex tasks, particularly those requiring long-horizon planning. *Future work* will focus on scaling this approach to such environments and exploring its integration with various RL algorithms, including model-based ones, to improve generalization in offline RL settings.

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
