# Supplementary Material for Synthetic Data is Sufficient for Zero-Shot Visual Generalization from Offline Data

**Ahmet H. Güzel**                                              *ahmet.guzel.23@ucl.ac.uk*
*University College London AI Centre*

**Ilija Bogunovic**                                              *i.bogunovic@ucl.ac.uk*
*University College London AI Centre*

**Jack Parker-Holder**                                          *j.parker-holder@ucl.ac.uk*
*University College London AI Centre*

**Reviewed on OpenReview:** *https://openreview.net/forum?id=gFmSFa408D*

## A  Code Repository

We will make our codebase and datasets available on `https://github.com/aguzel/augdiff_RL.git`. The code repository contains reference code bases in two folders: `V-D4RL` and `Procgen`. The README.md files in each sub-repository include detailed instructions for replicating the experimental results in this paper, training offline learning models, and creating additional datasets for use.

## B  Dataset Details

### B.0.1  Visual D4RL Environments

For our V-D4RL experiments, we utilized the expert offline dataset from Lu et al. (2023a) and conducted experiments on three environments:

- ***cheetah-run***: A planar bipedal agent rewarded for forward velocity.
- ***walker-walk***: A planar walker agent rewarded for upright posture and target velocity.
- ***humanoid-walk***: A humanoid agent with 21 joints rewarded for maintaining specific velocity.

We follow the V-D4RL approach, where learning from pixels is achieved by stacking three consecutive RGB images ($84 \times 84$) along the channel dimension to capture dynamic information such as velocity and acceleration. We utilize the expert dataset, with the baseline dataset size set at 100,000 samples per environment. The data collection policy is based on the Soft Actor-Critic (SAC) algorithm for proprioceptive states, as described by (Haarnoja et al., 2018).

**Visual D4RL Datasets:**

1. **50K Baseline**: A reduced dataset containing 50,000 samples, randomly sampled from the original 100,000 expert policy dataset, without any data augmentation.
2. **100K Upsampled**: An upsampled version of the 50,000 baseline dataset, increased to 100,000 samples using diffusion model-based upsampling.
3. **100K Augmented Baseline**: The original 100,000 dataset, augmented using the selected pixel-level augmentation techniques (rotation, color jittering, and background image overlay) to introduce additional diversity, without changing the dataset size.
4. **100K Augmented Upsampled**: An upsampled version of the 50,000 augmented baseline dataset, increased to 100,000 samples using diffusion model-based upsampling. This allows for a comparison between explicit augmentation and augmentation combined with diffusion upsampling.

**Evaluation Distraction Dataset:** The evaluation set, used for JS divergence analysis, is collected across all distraction levels as well as the original evaluation set. Data is gathered during training and combined across five random seeds. As a result, each environment's evaluation set for each testing difficulty level contains 256,000 samples. This larger size, compared to the original dataset, helps minimize variation in the test distribution.

We use the **Fixed Distraction Dataset (FDD):** provided by V-D4RL, which contains distractions fixed in the background and color of the agent—unlike the evaluation distracting dataset, where distractions change dynamically during evaluation. Previous attempts, as reported by (Lu et al., 2023a), to incorporate this dataset into training by combining various portions (25%, 50%, 75%, or even 100%) with the original dataset—without applying augmentation or our approach—have shown no improvement in generalization performance. Consequently, the authors highlighted an open challenge: *"How can we improve the generalization performance of offline model-free methods to unseen visual distractions?"*.

**5% Fixed Distraction Dataset (5% FDD):** To address this challenge, we incorporate a small portion of the fixed distraction dataset into our training data. Specifically, we use only 5% of the total fixed distraction dataset provided by the V-D4RL benchmark. In our reduced baseline of 50,000 datapoints, we replace 5,000 datapoints with samples from the fixed distraction dataset (combining low, moderate, and high levels), resulting in 45,000 original datapoints and 5,000 from the fixed distraction dataset. We believe that this addition introduces extra diversity to the original environment enabled by our two-step approach. This strategy can be considered a form of few-shot learning; however, since the 5% fixed distraction dataset is not part of the evaluation dataset, we refer to it as incorporating a **limited distraction exposure** in our training data. The **Fixed Distraction Dataset** is available exclusively for *cheetah-medium* and **cheetah-expert**. Therefore, we apply our approach solely to the **cheetah-expert** environment. Figure 1 shows the samples from the fixed distracting dataset for *cheetah-run.*

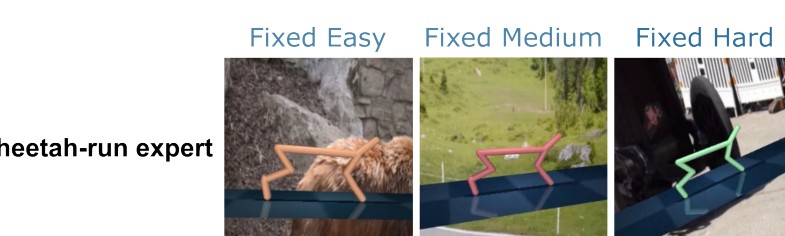

Figure 1: Sample screenshots from the V-D4RL cheetah-run fixed distracting dataset.

### B.0.2 Procgen Environments

We conducted experiments on three games from the Procgen suite:

- **CoinRun**: A platformer where the agent collects coins while avoiding obstacles.
- **Ninja**: An agent jumps across platforms, destroys bombs, and collects mushrooms.
- **Jumper**: A bunny navigates through multi-level environments, guided by a compass.

**Procgen Datasets for Experiments:**

1. **1M Baseline**: The original dataset containing 1,000,000 transitions without any data augmentation.
2. **2M Upsampled**: An upsampled version of the 1,000,000 baseline dataset, increased to 2,000,000 samples using diffusion model-based upsampling.
3. **1M Augmented Baseline**: The original 1,000,000 dataset, augmented using pixel-level augmentation techniques to increase diversity, without changing the dataset size.
4. **2M Augmented Upsampled**: An upsampled version of the 1,000,000 augmented baseline dataset.

For the Offline Procgen dataset, we use 1M expert dataset collected using Proximal Policy Optimization (PPO) Schulman et al. (2017). A single transition in Procgen consists of an observation (depicted as an

RGB image with dimensions 64x64x3), a discrete action (with a maximum action space of 15), a scalar reward (which may be either dense or sparse depending on the game), and a boolean value signifying the conclusion of the episode. Following the approach outlined by Mediratta et al. (2024), each Procgen game level is procedurally generated using a level seed, which is a non-negative integer. Levels in the range [0,200) are used to collect trajectories and train offline, levels [200,250) are utilized for hyperparameter tuning and model selection, and levels [250, ∞) are reserved for online evaluation of the agent's performance. We gather the **Evaluation Test Set** while training with 5 random seeds, employing a methodology similar to that used in our V-D4RL (Visual D4RL) study.

### B.0.3  Data Augmentation

To strengthen the robustness of our model to variations in visual inputs and ensure it captures key environment dynamics, we applied several data augmentation techniques to the initial dataset $\mathcal{D}_0$, inspired by the work of (Laskin et al., 2020). While we initially experimented with ten different augmentations, we found that using all of them simultaneously led to training instability. Through empirical analysis, we identified that *rotation*, *color jittering*, *color cutout*, and *background image overlay* were the most effective in improving generalization without compromising stability. We select applied augmentation randomly during the training. Below, we provide detailed descriptions of each of these augmentations. To further refine our augmentation strategy and reduce the time-consuming process of extensive tuning, we employed JS divergence analysis and visualization of the distributions of the upsampled and baseline datasets. This allowed us to assess alignment and diversity without repeated RL training, significantly streamlining the process. These augmentations, combined with diffusion-based upsampling, were instrumental in increasing data diversity and enhancing generalization.

The augmentation function applied to states $s$ and $s'$ uses independently sampled transformations from the same set of augmentations, with each transformation selected with equal probability. Within each state (e.g., a stack of frames), the same transformation is consistently applied across all images in the stack. This ensures that temporal and spatial relationships within the state are preserved, preventing disruption of critical structural information while still promoting diversity across states. This approach is consistent with the methodology used in RAD (Laskin et al., 2020), ensuring uniform exploration of the augmentation space while maintaining the structural integrity of stacked observations.

**Rotation:** We applied random rotations to the input images to make the model invariant to the orientation of objects within the environment. Each image was rotated by an angle randomly selected from a uniform distribution within a specified range, typically $[-\theta_{\max}, \theta_{\max}]$, where $\theta_{\max}$ is the maximum rotation angle which we set 90°. This augmentation helps the model generalize to different viewpoints and orientations it might encounter during deployment.

**Color Jittering:** To simulate variations in lighting conditions and color distributions, we employed color jittering. This technique involves randomly adjusting the brightness, contrast, saturation, and hue of the images. Specifically, we modified these properties by factors randomly sampled from uniform distributions around their original values. In our experiments, we adjusted the brightness in $[0.2, 0.6]$, contrast in $[0.2, 0.8]$, saturation in $[0.2, 0.8]$, and hue in $[0.1, 0.7]$. By introducing variability in the color space, the model learns to focus on features that are more robust to changes in illumination and color, improving its performance in diverse visual conditions.

**Color Cutout:** Color cutout is an augmentation technique where random rectangular regions of the image are occluded with a random color. Unlike standard cutout, which typically uses a fixed color (such as black or gray), color cutout replaces the occluded region with colors randomly sampled from the color space. Specifically, For each image, we selected one rectangle with dimensions up to 20% of the image's width and height. These regions were then filled with colors whose RGB values were uniformly sampled from $[0, 255]$. This augmentation forces the model to rely on contextual information from the visible parts of the image, improving its ability to handle occlusions and missing data in the input.

**Background Image Overlay:** To further broaden visual diversity and simulate different environmental conditions, we applied background image overlays. This technique involves blending the original images with randomly selected background images using random alpha channels. Specifically, we generated background

images using an image model (Rombach et al., 2022), and for each original image, we overlaid a background image with an alpha blending factor $\alpha$ randomly chosen from a uniform distribution in $[0.2, 0.5]$. The augmented image $I_{\text{aug}}$ is computed as:

$$I_{\text{aug}} = \alpha \times I_{\text{bg}} + (1 - \alpha) \times I_{\text{orig}},$$

where $I_{\text{bg}}$ is the background image and $I_{\text{orig}}$ is the original image. This augmentation exposes the model to a variety of background patterns and textures, helping it to generalize better to new environments where background elements may differ from those seen during training.

Figure 2 shows sample images from the Procgen environment, with similar techniques applied to the V-D4RL environment.

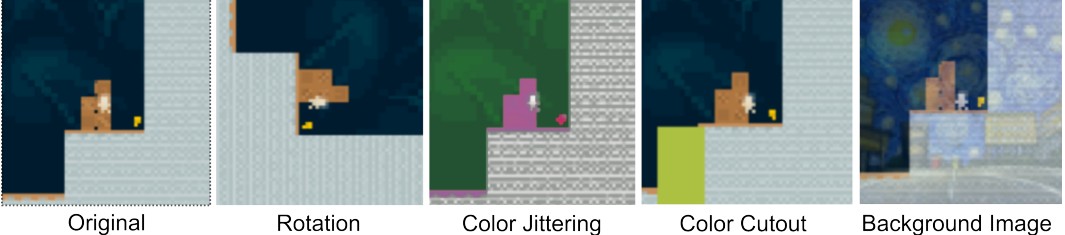

| Original | Rotation | Color Jittering | Color Cutout | Background Image |

Figure 2: Sample images of applied augmentation techniques for the Procgen Coinrun game.

## C    Algorithm Details and Hyperparameters

To ensure fair comparisons and validate the effectiveness of our method, we selected benchmark algorithms and datasets aligned with established offline RL benchmarks. Specifically, we used DrQ+BC and CQL for experiments on V-D4RL and Offline Procgen, respectively, following (Lu et al., 2023a) and (Mediratta et al., 2024). These algorithms were chosen due to their relevance to generalization challenges, as demonstrated in these papers, where DrQ+BC highlights issues in handling visual distractions and CQL underperforms in offline generalization tasks in Procgen, providing ideal test cases for evaluating our approach. While our focus is on demonstrating the algorithm-agnostic nature of our method, this section explains the selected algorithms, DrQ+BC and CQL, and the corresponding hyperparameters used in our experiments.

### C.1    DRQ+BC Algorithm

We utilize the DrQ+BC algorithm in our experiments, following the implementation described in Lu et al. (2023b). DrQ+BC builds upon DrQ-v2 (Yarats et al., 2021) by incorporating a behavioral cloning regularization term into the policy loss, similar to the approach used in TD3+BC (Fujimoto & Gu, 2021). As noted in Lu et al. (2023b), the base policy optimizer of DrQ-v2 shares similarities with TD3 (Fujimoto et al., 2018), which has been successfully adapted to offline settings from proprioceptive states by incorporating a regularizing behavioral cloning term into the policy loss. This modification results in the TD3+BC algorithm (Fujimoto & Gu, 2021).

Specifically, the policy objective becomes:

$$\pi = \arg\max_{\pi} \mathbb{E}_{(s,a)\sim D_{\text{env}}} \left[ \lambda Q(s, \pi(s)) - (\pi(s) - a)^2 \right], \tag{1}$$

and the loss function is expressed as:

$$\mathcal{L}_{\phi}(\mathcal{D}) = -\mathbb{E}_{s_t, a_t \sim \mathcal{D}} \left[ \lambda Q_{\theta}(h_t, a_t) - (\pi_{\phi}(h_t) - a_t)^2 \right]$$

Here, $\lambda$ is an adaptive normalization term computed over minibatches:

$$\lambda = \frac{\alpha}{\frac{1}{N} \sum_{(h_i, a_i)} |Q(h_i, a_i)|}$$

where $\lambda$ is a normalization term, $Q$ is the learned value function, and $\pi$ is the learned policy. The authors apply the same regularization technique to DrQ-v2 and refer to the resulting algorithm as **DrQ+BC**. The scalar $\alpha$ represents a behavioral cloning weight, which is fixed to 2.5 recommended by the authors.

The network architectures for the encoder, policy, and Q-function networks are consistent with those used in DrQ-v2 (Yarats et al., 2021). Specifically:

- **Encoder Network ($f_\xi$):** A convolutional neural network (CNN) with 4 layers, each with 32 channels, $3 \times 3$ kernels, and ReLU activations. The output is passed through a fully connected layer to produce a 50-dimensional latent vector.
- **Actor Network ($\pi_\phi$):** An MLP with two hidden layers of 1024 units each, using ReLU activations, outputting mean and standard deviation for a Gaussian policy.
- **Critic Network ($Q_\theta$):** An MLP with two hidden layers of 1024 units each, using ReLU activations, taking the concatenated state and action as input.

The hyperparameters used for the DrQ+BC algorithm in our experiments are summarized in Table 1.

Table 1: Hyperparameters for DrQ+BC Algorithm

| Parameter | Value |
|---|---|
| Batch size | 256 |
| Action repeat | 2 |
| Observation size | [84, 84] |
| Discount ($\gamma$) | 0.99 |
| Learning rate | $1 \times 10^{-4}$ |
| Optimizer | Adam |
| Agent training epochs | 256 |
| *n-step* returns | 3 |
| Exploration stddev. clip | 0.3 |
| Exploration stddev. schedule | linear(1.0, 0.1, 500000) |
| BC Weight ($\alpha$) | 2.5 |
| The number of training steps | 1,000,000 |

## C.2 CQL

We used the code base available at `https://github.com/facebookresearch/gen_dgrl` to implement CQL to our work. CQL (Conservative Q-Learning) is an offline reinforcement learning algorithm designed to penalize the overestimation of Q-values, thus encouraging conservative action selection. This approach helps avoid actions that have not been sufficiently explored in the dataset by regularizing the Q-function to lower the predicted values of out-of-distribution actions. CQL works especially well when offline data does not fully cover the state-action space. This makes policy evaluation and improvement more reliable (Kumar et al., 2020).

The Q-function objective reformulation:

$$\min_Q \alpha_{\text{CQL}} \mathbb{E}_{s \sim \mathcal{D}} \left[ \log \sum_a \exp(Q(s,a)) - \mathbb{E}_{a \sim \hat{\pi}_\beta(a|s)}[Q(s,a)] \right] + \frac{1}{2} \mathbb{E}_{s,a,s' \sim \mathcal{D}} \left[ \left( Q - \hat{\mathcal{B}}^{\pi_k} Q^k \right)^2 \right].$$

In this formulation, $\alpha_{\mathrm{CQL}}$ is the trade-off parameter, $\hat{\pi}_\beta$ is the empirical behavioral policy, and $\hat{\mathcal{B}}^{\pi_k}$ denotes the empirical Bellman operator that updates a single sample. We approximate this by taking gradient steps and sampling actions within the defined bounds.

**Network architecture:** The network architecture is adopted from (Mediratta et al., 2024), which is detailed below.

- **Encoder Network**: ResNet-based convolutional neural network with approximately 1 million parameters (He et al., 2015).
- **Q-Function Network**: The encoder outputs a latent vector that is passed through a fully connected layer with 256 units and ReLU activation, followed by an output layer producing Q-values for each of the 15 discrete actions.

The hyperparameters used for the CQL algorithm in our Procgen experiments are listed in Table 2.

Table 2: Hyperparameters for CQL Algorithm

| Parameter | Value |
|---|---|
| Batch size | 256 |
| Learning rate (Q-function) | $5 \times 10^{-4}$ |
| Optimizer | Adam |
| Epochs | 1000 |
| Q-Function Network Hidden Size | 256 |
| Target Network Update Frequency | 1000 |
| $\tau$ | 0.99 |
| $\alpha$ | 4.0 |

### C.3 Elucidated Diffusion Model

In our work, we employ the Elucidated Diffusion Model (EDM) proposed by Karras et al. Karras et al. (2022). The denoising network $D_\theta$ is parameterized as a multilayer perceptron (MLP) with skip connections from the previous layer, following the architecture described by Tolstikhin et al. Tolstikhin et al. (2021). Each layer of the network is defined as:

$$x_{L+1} = \mathrm{Linear}(\mathrm{Activation}(x_L)) + x_L, \tag{2}$$

where $x_L$ is the input to layer $L$, and the activation function is typically ReLU.

The hyperparameters used for the denoising network are listed in Table 3. To encode the noise level of the diffusion process, we utilize a Random Fourier Feature (RFF) embedding as introduced by Tancik et al. Tancik et al. (2020). The base network has a width of 1024 neurons per layer and a depth of 6 layers, resulting in approximately 6 million parameters.

We adjust the batch size during training based on the size of the dataset. For online training and offline datasets with fewer than 1 million samples (e.g., medium-replay datasets), we use a batch size of 256. For larger datasets, we increase the batch size to 1024. For *V-D4RL*, we employ uniform hyperparameters for both augmented and baseline training to maintain consistency. Likewise, we implement the identical principle to the *Procgen* dataset. The hyperparameters are listed in Table 3.

Additionally, we conducted a similar ablation study to that performed in SynthER (Lu et al., 2023b) and observed results consistent with those reported by the authors. To avoid redundancy, we did not include an exhaustive presentation of these ablations in our paper. For a detailed analysis, we refer readers to the SynthER supplementary material (Section C and Section F), which provides comprehensive insights into their findings.

Table 3: Default Hyperparameters for the Residual MLP Denoiser.

| Parameter | Value(s) |
|---|---|
| Number of layers | 6 |
| Width | 1024 |
| Batch size | 1024 |
| RFF dimension | 16 |
| Activation function | ReLU |
| Optimizer | Adam |
| Learning rate | $3 \times 10^{-4}$ |
| Learning rate schedule | Cosine annealing |
| Number of training steps | 100,000 |

For the diffusion sampling process, we use the stochastic SDE sampler from Karras et al. Karras et al. (2022) with the default hyperparameters used for ImageNet, as shown in Table 4. We employ a higher number of diffusion timesteps, set to 128, to improve sample fidelity. The implementation is based on the publicly available code at `https://github.com/lucidrains/denoising-diffusion-pytorch`, which is released under the Apache License.

Table 4: Default Hyperparameters for the ImageNet-64 EDM.

| Parameter | Value |
|---|---|
| Number of diffusion steps | 128 |
| $\sigma_{\min}$ | 0.002 |
| $\sigma_{\max}$ | 80 |
| $S_{\text{churn}}$ | 80 |
| $S_{t_{\min}}$ | 0.05 |
| $S_{t_{\max}}$ | 50 |
| $S_{\text{noise}}$ | 1.003 |

### C.3.1 Ablation Study for Latent Space Dimension

To complete the ablation studies from the original SynthER approach, we performed an additional analysis focusing on latent space dimension size. This aspect was not covered in the original SynthER work. The dimensionality settings align with those used in offline RL algorithms, as detailed in Sections C.1 and C.2. This study extends the understanding of the role of latent space dimensionality in generalization performance.

| Environment | Method | Latent Dimension Size | Normalized Generalization Performance |
|---|---|---|---|
| *V-D4RL Averaged* | Ours | 32 | 0.92 |
| | Ours | $64(default)$ | 1.0 |
| | Ours | 128 | 0.96 |
| | Ours | 256 | 0.79 |
| *Procgen Averaged* | Ours | 64 | 0.94 |
| | Ours | $100(default)$ | 1.0 |
| | Ours | 256 | 0.92 |
| | Ours | 512 | 0.68 |

From our results, we observe that reducing the latent space dimension compromises the model's ability to capture the underlying distribution of the data effectively. In V-D4RL, for instance, dimensions smaller than the default value result in poor performance as the reduced representation fails to encapsulate the necessary data diversity. On the other hand, increasing the latent space dimension beyond the default introduces

challenges for the diffusion model, requiring larger denoising networks. This not only increases computational costs but also risks overfitting to the training data. For the largest latent dimension, the diffusion model struggles to learn an aligned distribution of augmented V-D4RL, further degrading performance.

A comparable tendency is evident in Procgen. Insufficiently small latent dimensions fail to encapsulate the data's diversity, resulting in inferior performance. Larger dimensions increase processing demands and diminish the learning capability of the diffusion model.

### C.4 Computational Cost Analysis

We evaluate the computational time required for our proposed approach compared to the baseline (without augmentation and upsampling) on an NVIDIA 4090 GPU.

In addition to the baseline runs for both benchmarks, our method introduces two supplementary computational overheads: one for the augmentation process and the other for the upsampling procedure. Applying the initial augmentation step to images increases the overall computational cost, which extends the runtime. However, because our method leverages latent space upsampling, the dimensionality reduction mitigates some of the computing costs compared to directly operating on high-dimensional images.

The Table C.4 below summarizes the computational costs for a single seed run, highlighting the baseline and two variants of our method.

| Environment | Method | Aug. | Upsampling | Runtime (hours) |
|---|---|---|---|---|
| V-D4RL | 100K Baseline | ✗ | ✗ | 3.49 |
| | 100K Augmented + Upsampled | ✓ | ✓ | 7.50 |
| Procgen | 1M Baseline | ✗ | ✗ | 0.20 |
| | 2M Augmented + Upsampled | ✓ | ✓ | 0.37 |

Table 5: Computational cost analysis for different methods across V-D4RL and Procgen environments.