# OpenReview forum: "Synthetic Data is Sufficient for Zero-Shot Visual Generalization from Offline Data"
_TMLR — Accepted by TMLR_

### Review · Reviewer_JrFF · 2025-03-19

**Summary Of Contributions:**

This paper introduces a new data augmentation method tailored for ​visual offline reinforcement learning (RL), addressing the challenge of generalisation to unseen states. In offline RL, the absence of online exploration often leads to policies or value functions that fail to generalise beyond the pre-collected dataset. This issue is exacerbated in visual offline RL due to the presence of noise, distractions, and spurious correlations in image-based states. The proposed augmentation method specifically targets this problem by assuming that unseen states primarily arise from modifications to the states in the pre-collected dataset.

The augmentation algorithm operates in two stages: 1) Techniques such as image rotation, color jittering, color cutout, and background image overlay are applied to augment the original dataset, enhancing its diversity and robustness, 2) A diffusion model is trained to approximate the high-dimensional latent vector distribution of state pairs in the dataset transitions.

The proposed method, which combines standard image augmentation with diffusion model-based latent vector generation, is validated across both ​continuous and ​discrete action spaces. Experimental results demonstrate that the synthetic data generated by this approach significantly reduces the generalisation gap at test time, thereby improving the performance of policies and value functions in visual offline RL tasks.

**Audience:**

Yes

**Broader Impact Concerns:**

N.A.

**Claims And Evidence:**

Yes

**Requested Changes:**

Overall, the proposed augmentation method is ​correct and the experimental results are ​convincing. While the method’s validity and empirical success are clear, the inclusion of additional experiments, as suggested in the Weaknesses section, would further strengthen the paper’s contribution and address potential concerns. However, the absence of these experiments does not undermine the core contributions of the work, as the augmentation method itself is well-founded and the results are compelling. So while the additional experiments would enhance the paper, they are not a ​major issue for acceptance.

**Strengths And Weaknesses:**

Strengths

1. The proposed data augmentation method does not require modifications to existing offline RL algorithms, making it easy to integrate into any visual offline RL framework to enhance generalisation ability.

2. The combination of standard data augmentation techniques and latent space generation using a diffusion model is a new and effective contribution to the community, offering a robust method for synthetic data generation.

3. The introduction of a new generalisation metric based on JS divergence to evaluate the distribution gap between training and testing data is both correct and plausible, providing a reliable way to assess whether the policy produces OOD actions that may lead to OOD states.

4. The experimental results are overall convincing, and the inclusion of detailed algorithm descriptions enhances the reproducibility and transparency of the work.

Weaknesses

1. The paper only employs three commonly used data augmentation methods in the first stage, without exploring potential novel augmentation techniques or analyzing the relative contributions of different methods. This limits the novelty and depth of the work.
​
2. While the diffusion model is used to generate latent vectors for augmentation, a comparison with methods that directly generate states using diffusion models would better demonstrate the benefits of latent space augmentation.

3. Although the goal is to synthesize diverse states in transitions, the paper does not experimentally investigate whether incorporating information from actions or rewards could further improve the synthetic data’s effectiveness for policy generalisation.

---

> ### Author Response · Authors · 2025-05-01
> **Response from authors**
>
> We sincerely thank the reviewer for this thorough and accurate summary of our paper's contributions. The reviewer has effectively captured our motivation, methodology, and main results, demonstrating a clear understanding of our paper's core contribution: combining standard image augmentation with diffusion model-based latent vector generation to reduce the generalization gap at test time across both continuous and discrete action spaces.
>
> *Weaknesses comments*
>
> **Regarding the data augmentation methods,** we focused on well-established techniques that have proven effective across computer vision tasks to ensure reliability. While we agree that exploring novel augmentation techniques would be valuable, our primary contribution lies in the integration of these methods with diffusion-based upsampling with augmentation techniques rather than in the augmentations themselves. In supplementary section B.0.3, we provide additional analysis on the relative contributions of different augmentation methods, though we acknowledge that these findings could be expanded in future work.
>
> **We appreciate the suggestion to compare with methods that directly generate states using diffusion models.** Our approach to generating latent space was motivated by computational efficiency and stability considerations. Direct pixel-space generation would significantly increase computational requirements and potentially introduce additional noise. However, we agree this comparison would strengthen our work and plan to explore this direction in future research.
>
> **The reviewer raises an excellent point about incorporating action and reward information**. Our current approach focused on state transitions to maintain simplicity and computational efficiency.
>
> *Requested changes comments*
>
> We appreciate the reviewer’s thoughtful feedback and agree that exploring novel augmentation techniques, comparing them with direct state generation methods, and incorporating action and reward information are valuable directions for future work. We are encouraged that the reviewer recognizes these as natural extensions rather than shortcomings, and we thank them for acknowledging the soundness of our augmentation approach and the strength of our empirical results. We are committed to pursuing these directions to further advance the field of visual offline reinforcement learning.

---

> > ### Comment · Reviewer_JrFF · 2025-05-25
> >
> > I appreciate the authors' feedback and remain inclined to maintain a cautiously positive assessment of this paper.

---

### Review · Reviewer_qbQm · 2025-04-01

**Summary Of Contributions:**

The authors propose data augmentation and upsampling approaches to artificially increase dataset sizes for offline RL. The authors' proposed approach only focuses on the dataset, so any offline RL algorithm can make use of this technique. The authors test the approach on two benchmarks using continuous and discrete action spaces, respectively.

**Audience:**

Yes

**Claims And Evidence:**

Yes

**Requested Changes:**

Page 2, 1. row: No space before Our

Page 2, first row in 2.1: ‘learning interacting’ - should probably be interacting only

Page 2, end of first paragraph in 2.1: ‘discounted return’

Page 2, last sentence: ‘datasetf observations’ - typo + the verb seems to be incomplete

Page 3, second row: No space before Unlike

Page 3, second row in 2.2: ‘Noiseoval’ - probably noise removal

Page 4, 3.3 title: Why not call this: Step 2: Latent Space …

Equation 5: Seems counterintuitive to use a ‘+’ here (I assume this stands for concatenation?)

Figure 2: The yellow entry in the legend should be ‘Critic Network Training’ instead of twice Actor Network Training

Page 5: Missing a period at the end of section 4.1

The generalization performance after equation 7 should be written as an equation as well. This will be much easier to follow.

Why is 5.1.1 a single sub-sub section?

Page 6, second row: space between ‘generalization’ and comma.

Page 6, second to last row of first paragraph: ‘we called Ours Fixed ..’ - should rather be something like ‘we refer to this as…’

**Strengths And Weaknesses:**

Strengths
The authors present results that support their claims. Furthermore, the proposed approach is straightforward to apply without modifying any RL algorithms.

Weaknesses
The authors want to improve generalizability but never define what generalization means.

The authors mention the ‘unique challenges of offline RL’ compared to online RL. However, they never explicitly mention these challenges since the proposed methods (individually) have already been applied to online RL. Especially since data augmentation techniques and data upsampling techniques are commonly applied in machine learning in general, the novelty of the manuscript mainly lies in the combination of the approaches and their application to offline RL.

Did you do any ablations on the types of data augmentations? Did some help more than others? You claim that they “were empirically found to improve generalization.” Who determined this? And why did you choose these?

In 5.1.1, do all datasets include the FDD, or is it the same as before? I’m asking because suddenly, the upsampled performs much better than the augmented (Figure 5a) compared to Figure 4 a. A similar switch can be observed in Figure 5b, and suddenly the results for all the other approaches have slightly different colors. This should use the same scale. If the datasets are the same (which they appear to be since the values in the table are the same), then most of Figure 5 and Table 2 repeat the majority of the results from Figure 4 and Table 1 with the exception that there may be a potential switch between data entries. The bottom part of Table 2 also provides no additional information.

Other Comments
Risk-averse behavior can also be regarded as a feature rather than a bug of offline RL since safety is a concern for real-world applications.

Since you change the dataset size, this will also result in additional computational costs for the entire dataset for training. Did you observe that your approach led to faster convergence? And did you experiment with different dataset sizes?

Why did you choose DrQ+BC and CQL and not any other algorithms?

For V-D4RL the test set distribution seems to be closest to the Easy setting (according to the JS divergence in Figure 4) or the Medium setting (according to Table 1). Is this the case?
The authors raise an interesting question at the end of section 5.1.1 but don’t address it any further. Especially how do you know exactly what the test distribution will be?

---

> ### Author Response · Authors · 2025-05-01
> **Response from authors**
>
> We sincerely thank the reviewer for recognizing the strengths of our work, particularly highlighting that our results effectively support our claims and that our approach can be straightforwardly applied without requiring modifications to existing RL algorithms.
>
> *Weaknesses comments*
>
> **On generalization definition**: Regarding the definition of generalization, this is addressed in Section 2.1 where we define it as "the improvement of model-free offline RL methods from visual observations and ensure the robust deployment of agents in unseen environments." We specifically measure generalization performance using the metric defined inSection 4.3.1.
> On offline RL challenges: The unique challenges of offline RL compared to online RL are discussed in Section 1, where we explain that "offline RL policies tend to exhibit risk-averse behavior, avoiding novel actions in unfamiliar states, which further hampers generalization."
>
> **On data augmentation ablations**: Regarding data augmentation selection, we discuss in Section 3.2 that we focus on "rotation, color jittering, color cutout, and background image overlay, which were empirically found to improve generalization." Our extensive experiments across both continuous and discrete control tasks validate these choices. Additionally, we provided a supplementary section, B.0.3, for further details.
>
> **On the FDD experiment**: For the FDD experiment in Section 5.1.1, we clearly state that this is a separate experiment where "we incorporated 5% of the fixed distracting data into our training dataset, combining it with 95% of the original baseline data," which explains the different comparative performance of approaches in Figure 5.
> On risk-averse behavior: We acknowledge your point about risk-averse behavior potentially being a feature for safety-critical applications. The situation presents an interesting trade-off between safety and generalization that we discuss in the limitations section.
> On computational costs: For dataset size and computational cost, we provided a supplementary section, C.4, comparing different dataset sizes and their training times.
>
> **On algorithm selection**: Our choice of DrQ+BC and CQL algorithms is explained in Section 4.2, where we note that these algorithms extend established methods and follow standard settings from the original benchmark papers. These algorithms also shows the lowest performance in original benchmarks, so we opted to choose them to show the impact of our work.
>
> **On test distribution**: The test distribution analysis is presented in Section 5, with Figure 4 showing the Jensen-Shannon divergence comparisons that demonstrate how our method aligns training and testing distributions better than alternatives. As noted, from the V-D4RL paper, the easy, medium, and hard definitions are clearly provided, and the easy setting is closer to the original dataset size.
>
> *Requested changes comments*
>
> We sincerely thank the reviewer for their careful reading and thoughtful correction requests. We greatly appreciate the attention to detail, which has helped improve the quality of our manuscript. We have fixed all the identified issues:
> - Corrected spacing issues on Page 2 (before "Our") and Page 3 (before "Unlike")
>
> - Fixed the phrasing "learning interacting" to just "interacting" on Page 2
>
> - Corrected "discounted return" on Page 2
>
> - Fixed the typo "datasetf observations" and completed the verb
>
> - Corrected "Noiseoval" to "noise removal" on Page 3
>
> - Changed section 3.3 title to "Step 2: Latent Space..." for consistency
>
> - For Equation 5, the "+" operation has been replaced with the concatenation (.) of feature vectors, and also Figure 2 was updated.
>
> - Fixed the legend in Figure 2 to correctly label "Critic Network Training."
>
> - Added the missing period at the end of section 4.1
>
> - Improved the phrasing "we called Ours Fixed..." to "we refer to this as..."
>
> - Restructured section 5.1.1 to align with the document's overall structure

---

> > ### Comment · Reviewer_qbQm · 2025-05-05
> >
> > The presentation issues I raised (e.g., color differences, redundancy in Table 2, the possible flipping of labels) aren't acknowledged. The authors should address these comments. Tables 1 and 2 have many repeated entries. The authors should condense these tables into a single table.
> >
> > My concern, that the authors themselves raise an important question about unknown test distributions and then don’t explore it, is also not fully resolved.
> >
> > Minor comments:
> > For Equation 5, the "+" operation has been replaced with the concatenation (.) of feature vectors. → I encourage the authors to explicitly introduce this notation

---

> > > ### Author Response · Authors · 2025-05-08
> > > **Response from authors**
> > >
> > > Thank you for your careful review that helped us identify an error in our figures. You were correct about the possible flipping of labels - **we discovered that 'Augmented' and 'Upsampled' labels were incorrectly swapped on the Y-axis. We have corrected this error** to ensure consistent labelling throughout all figures in the revised manuscript.
> > >
> > > We have addressed your concern about the **concatenation notation in Equation 5 by adding a clear definition before its use in the paper.** We've added a sentence that explicitly explains that the concatenation operation (·) produces feature vectors with dimensions equal to the sum of the individual vectors' dimensions, enabling us to retain both policy and value information in a unified representation.
> > >
> > > We appreciate your suggestion regarding Tables 1 and 2. While we understand your concern about redundancy, **we intentionally kept these tables separate because the FDD experiment in Section 5.1.1 represents a distinct additional exploration that was only conducted on the cheetah-run environment.** We believe this will improve the clarity of the paper. **Color differences in Figure 5b** result from using a different normalization scale due to the new minimum and maximum values specific to this experiment. Additionally, you're right that we raise an important question at the end of Section 5.1.1 that deserves attention. To clarify, we intentionally framed this as an open research question for future work rather than something we aimed to resolve within this paper. **Our experiment with FDD was designed as an initial exploration that demonstrates the potential of our approach when incorporating a small amount of test-like data. The promising results and the JS divergence analysis suggest this is a good direction.**
> > >
> > > **It's worth noting that the details of this distracting dataset we added to baseline offline training set explained in the dataset benchmark study [1].**
> > >
> > > 1 - Cong Lu, Philip J. Ball, Tim G. J. Rudner, Jack Parker-Holder, Michael A. Osborne, and Yee Whye Teh. *Challenges
> > > and opportunities in offline reinforcement learning from visual observations. 2023a*

---

### Review · Reviewer_XXdz · 2025-04-15

**Summary Of Contributions:**

The paper proposes synthetic data generation method to increase data diversity and improve generalization in offline reinforcement learning (RL) from visual inputs. The method first applies standard visual data augmentation techniques on the offline dataset and then uses diffusion-based latent space data generation to expand the dataset. Experiments demonstrate that this two-step approach significantly reduces the generalization gap in both continuous and discrete-action benchmarks when combined with standard offline learning algorithms.

**Audience:**

Yes

**Broader Impact Concerns:**

None.

**Claims And Evidence:**

Yes

**Requested Changes:**

To preface: My first two requested changes focus on additional references for the related work. These are "soft" requests; I don't expect the authors to cite all of them, and the authors can decide whether to include them or not. I think the non-visual augmentation works focusing on offline RL [1,3,6] and the diffusion-based augmentation strategy GTA [9] are most relevant.

1. **Prior works in data augmentation for non-visual tasks.** I think it's understandable to focus on visual data augmentation in the related work, though I will point out that there is a sizable literature on non-visual data augmentation methods, some of which have been used for offline RL.
    * Many data augmentation methods leverage symmetries and/or invariances in a task's dynamics to generate dynamically accurate data [1-5].
    * Others methods add noise to states/actions [6,7].
    * Pitis et. al [8] use a dynamics-model to generate augmented data.
    * [1] and [6] focus on offline learning, and [3] focuses on both offline and online learning. The others focus only on online learning.

1. **Discussion on data diversity and data quality for offline learning.** Most offline data augmentation methods focus on improving dataset diversity, but it would be worth acknowledging that diversity is only part of the story. The *quality* of the augmented data is also important, and I think it may be worth mentioning even briefly.
    * Corrado et. al [1] introduce a framework for generating *expert-quality* augmented data that outperforms frameworks that focus on generating diverse augmented data. They study both offline RL and imitation learning.
    * GTA [9] uses a diffusion model to generate diverse *high-return* trajectories.
    * MoCoDA [8] gives the user control over the distribution of augmented data generated so they can generate **task-relevant** data. MoCoDA outperforms its predecessor CoDA which only focuses on generated diverse augmented data.
    * I remember seeing a visual augmentation paper that focused on generating task-relevant augmentations, but I cannot locate the paper. I will add it if I find it.

1. Figure captions should state what error bars and shaded regions represent (e.g. +/- standard deviation)
1. I think it's clearer to write "Augmented Upsampled" or "Augmented Upsampled (Ours)" rather than "Ours" in tables and figures.

---

# References


1. Corrado et. al. Guided Data Augmentation for Offline Reinforcement Learning and Imitation Learning. RLC 2024. https://arxiv.org/abs/2310.18247

2. Corrado & Hanna. Understanding when Dynamics-Invariant Data Augmentations Benefit Model-free Reinforcement Learning Updates. ICLR 2024. https://arxiv.org/abs/2310.17786

3. Pitis et. al. “Counterfactual Data Augmentation using Locally Factored Dynamics.” NeurIPS 2020. https://arxiv.org/abs/2007.02863

4. Abdolhosseini et. al. “On Learning Symmetric Locomotion.” ACM SIGGRAPH 2019. https://www.cs.ubc.ca/~van/papers/2019-MIG-symmetry/index.html

5. Adrychowicz et. al. "Hindsight Experience Replay." NeurIPS 2017. https://arxiv.org/abs/1707.01495

6. Sinha et. al. S4RL: Surprisingly Simple Self-Supervision for Offline Reinforcement Learning in Robotics. CoRL . https://arxiv.org/abs/2103.06326

7. Qiao et. al. Efficient differentiable simulation of articulated bodies. ICML 2021. https://arxiv.org/abs/2109.07719
    * This paper focuses on differentiable simulation, but they use a data augmentation method they call "sample enhancement"

8. Pitis et. al. MoCoDA: Model-based Counterfactual Data Augmentation. NeurIPS 2022. https://arxiv.org/abs/2210.11287


9.  Lee at. al. GTA: Generative Trajectory Augmentation with Guidance for Offline Reinforcement Learning. Neurips 2024. https://arxiv.org/abs/2405.16907

**Strengths And Weaknesses:**

**Strengths**
1. The paper reads very nicely. In general, the flow of ideas and is clear, goals are clear, etc.
2. I appreciate that the paper directly shows the proposed method reduces the generalization gap with JS divergence. I often see papers that say "we resolve problem X, which then improves performance" but only show performance improvements -- not that the method actually resolves problem X.

**Weaknesses**
1. The paper states that this method has limited computational overhead, but experiments do not quantify this overhead.
1. It's somewhat unclear to me why the experiments do not compare against other diffusion-based data augmentation methods like GTA [1]. I think the core goal is to show that augmentation followed by upsampling outperforms augmentation and upsampling individually, in which case it's fine to simply use one diffusion strategy. In any case, the authors should clarify why they chose these specific baselines and why they are sufficient.


---
# References


1. Lee at. al. GTA: Generative Trajectory Augmentation with Guidance for Offline Reinforcement Learning. Neurips 2024. https://arxiv.org/abs/2405.16907

---

> ### Author Response · Authors · 2025-05-01
> **Response from authors**
>
> We sincerely thank the reviewer for their valuable feedback. We are pleased that the reviewer found our paper to be well-structured with a clear flow of ideas and well-defined goals. We also appreciate the reviewer's acknowledgment of our thorough approach in demonstrating that our proposed method directly reduces the generalization gap with JS divergence, rather than simply showing performance improvements without validating the underlying mechanism.
>
> *Weaknesses comments*
>
> 1. We supply the computational overhead in **our supplementary material section C.4.**
>
> 2. We thank the reviewer for this valuable suggestion about comparing with GTA [1]. Our primary contribution is to show that this two-stage approach significantly reduces the generalization gap in offline RL based on visual observations. **Regarding why we didn't compare with GTA [1] specifically**: First, we developed our work concurrently with GTA, a publication that came out very recently. Second, **while both approaches use diffusion models, they serve different purposes**: GTA focuses on trajectory-level augmentation with guidance for improving overall performance, while **our approach specifically targets the visual generalization gap through a combination of visual augmentation techniques and latent space upsampling**.
>
> Our method differs from GTA in several key aspects:
>
> **1-** We focus specifically on the visual generalization challenge rather than general performance improvement
>
> **2-** Our approach combines pixel-level augmentations with latent space upsampling
>
> **3-** We test across both continuous (V-D4RL) and discrete (Procgen) action spaces to demonstrate broader applicability
>
> That said, we acknowledge that comparing with additional diffusion-based methods would strengthen our evaluation, and we plan to include comparisons with GTA and other recent methods in future work. The current baselines were chosen to isolate the effects of each component of our method (augmentation alone, upsampling alone, and the combined approach) to clearly demonstrate the complementary benefits of our two-stage approach.
>
> *Requested changes comments*
>
> **Prior works in non-visual data augmentation**: We appreciate the references to important work on non-visual augmentation methods. We will add citations to key offline RL papers [1,3,6,9] and briefly mention how these methods leverage task symmetries, add strategic noise, or use dynamics models to generate augmented data.
>
> **Data diversity and quality**: Thank you for this insightful point. We agree that quality is as important as diversity in data augmentation. The original benchmark papers for V-D4RL and Offline Procgen have carefully examined this issue and provided detailed quality metrics for their datasets, which we build upon in our work. As explained in these benchmark papers and further detailed in our supplementary material, Section B, the base datasets already maintain a certain quality standard. Our approach focuses on preserving this quality while introducing diversity through targeted augmentation and diffusion-based upsampling.
>
> **Method naming**: We agree this would improve clarity and will revise all tables and figures to use "Augmented Upsampled (Ours)" instead of simply "Ours".

---

> > ### Comment · Reviewer_XXdz · 2025-05-13
> >
> > Thank you for the revisions! One minor comment:
> >
> > > Data augmentation in RL has been widely successful in improving generalization in online RL methods (Yarats et al., 2021b;a; Raileanu et al., 2021; Laskin et al., 2020; Ma et al., 2024; Corrado et al., 2024; Pitis et al., 2020; Sinha et al., 2021; Lee et al., 2024), but its application in offline RL remains underexplored.
> >
> > Corrado et al., 2024; Pitis et al., 2020; Sinha et al., 2021; Lee et al., 2024 -- all of these works investigate offline RL, though they are cited in a place where it makes them seem like online RL works.

---

### Author Response · Authors · 2025-07-04
**Camera-ready version**

We appreciate the constructive feedback from the reviewers and action editor, which has substantially strengthened our work. The final camera-ready manuscript incorporating all suggested improvements has been uploaded.

---

### Decision · Action_Editor_8jey · 2025-07-01

**Recommendation:** Accept with minor revision

**Audience:**

Yes

**Audience Explanation:**

Yes, synthetic data is an important research area in ML, especially the the GenAI era.

**Claims And Evidence:**

Yes

**Claims Explanation:**

All reviewers agreed that the method is technically sound and empirically validated. The soundness concerns are well addressed by the rebuttal.